# Mobile-HR: An Ophthalmologic-Based Classification System for Diagnosis of Hypertensive Retinopathy Using Optimized MobileNet Architecture

**DOI:** 10.3390/diagnostics13081439

**Published:** 2023-04-17

**Authors:** Muhammad Zaheer Sajid, Imran Qureshi, Qaisar Abbas, Mubarak Albathan, Kashif Shaheed, Ayman Youssef, Sehrish Ferdous, Ayyaz Hussain

**Affiliations:** 1Department of Computer Software Engineering, MCS, National University of Science and Technology, Islamabad 44000, Pakistan; 2College of Computer and Information Sciences, Imam Mohammad Ibn Saud Islamic University (IMSIU), Riyadh 11432, Saudi Arabia; iqureshi@imamu.edu.sa; 3Department of Multimedia Systems, Faculty of Electronics, Telecommunication and Informatics, Gdansk University of Technology, 80-233 Gdansk, Poland; kashif.shaheed@pg.edu.pl; 4Department of Computers and Systems, Electronics Research Institute, Cairo 12622, Egypt; 5Department of Software Engineering, National University of Modern Languages, Rawalpindi 44000, Pakistan; sehrish.ferdous@numl.edu.pk; 6Department of Computer Science, Quaid-i-Azam University, Islamabad 44000, Pakistan; ayyaz.hussain@qau.edu.pk

**Keywords:** computer-aided diagnosis, hypertensive retinopathy, retinal fundus images, deep learning, transfer learning, convolutional neural network, MobileNet, DenseNet

## Abstract

Hypertensive retinopathy (HR) is a serious eye disease that causes the retinal arteries to change. This change is mainly due to the fact of high blood pressure. Cotton wool patches, bleeding in the retina, and retinal artery constriction are affected lesions of HR symptoms. An ophthalmologist often makes the diagnosis of eye-related diseases by analyzing fundus images to identify the stages and symptoms of HR. The likelihood of vision loss can significantly decrease the initial detection of HR. In the past, a few computer-aided diagnostics (CADx) systems were developed to automatically detect HR eye-related diseases using machine learning (ML) and deep learning (DL) techniques. Compared to ML methods, the CADx systems use DL techniques that require the setting of hyperparameters, domain expert knowledge, a huge training dataset, and a high learning rate. Those CADx systems have shown to be good for automating the extraction of complex features, but they cause problems with class imbalance and overfitting. By ignoring the issues of a small dataset of HR, a high level of computational complexity, and the lack of lightweight feature descriptors, state-of-the-art efforts depend on performance enhancement. In this study, a pretrained transfer learning (TL)-based MobileNet architecture is developed by integrating dense blocks to optimize the network for the diagnosis of HR eye-related disease. We developed a lightweight HR-related eye disease diagnosis system, known as Mobile-HR, by integrating a pretrained model and dense blocks. To increase the size of the training and test datasets, we applied a data augmentation technique. The outcomes of the experiments show that the suggested approach was outperformed in many cases. This Mobile-HR system achieved an accuracy of 99% and an F1 score of 0.99 on different datasets. The results were verified by an expert ophthalmologist. These results indicate that the Mobile-HR CADx model produces positive outcomes and outperforms state-of-the-art HR systems in terms of accuracy.

## 1. Introduction

Hypertensive retinopathy is the most common cause of retinal disease (HR). According to reports from the World Health Organization (WHO), most of the increase in the number of people with hypertension has occurred in low- and middle-income countries, increasing from 594 million in 1975 to 1.13 billion in 2015. The prevalence of existing conditions that enhance the possibility of developing hypertension is primarily responsible for this trend. A total of 1.56 billion people are expected to have hypertension by the year 2025. In addition, over 66% of people with hypertension reside in poor or underdeveloped countries, which makes the condition worse due to the lack of resources for healthcare that may be used in its diagnosis, monitoring, and treatment [1].

In practice, the ophthalmoscopic features of an HR examination show the effects on arteriolar constriction, arteriovenous nicking, vascular wall changes, flame-shaped hemorrhages, cotton wool spots, yellow hard exudates, and optic disk edema. Hypertension-induced ocular damage mainly includes choroidopathy, optic neuropathy, and hypertensive retinopathy [2]. Hypertensive retinopathy (HR) is an important disease to classify because it can cause vision loss. HR might also cause heart disease, which can be fatal. Thus, it has been recognized as posing a severe threat to general human health worldwide. If hypertension is detected and treated early, the risk of HR may be decreased. The early stages of HR are difficult to identify, since there are not enough experienced ophthalmologists or advanced imaging technologies that can classify the disease at this stage [3]. HR symptoms induce nicking of the retina, arteriolar narrowing, and arteriovenous narrowing [4]. Cotton wool spots, hemorrhages, papilledema, microaneurysms, and optical nerve and retinal edema are further notable signs of an eye condition associated with HR. Previous literature suggests that approximately 10% of adults without diabetes have modest signs of HR [5]. Fundus images captured by an optical device can reveal retinal microvascular abnormalities, according to recent studies. Due to the fact of its low cost, ease of use, and ability to accurately portray many clinical lesion structures in its fundus images, this fundus camera is frequently used to safely evaluate HR patients [6]. According to several research studies [7], mobile-based AI can assist in the detection of HR. Because mobile devices have lower memory and processing capacities, most of the significant research effort focuses on employing designs that are bulky and computationally costly. This paper presents a Mobile-HR system that classifies HR into binary classes using a multilayer MobileNet architecture that integrates dense blocks. A dense block may create fewer convolution kernels and more feature maps, fully utilize the output feature maps of the preceding convolution layers, and realize the recurrent usage of features. To make MobileNet’s parameters and calculations even more manageable for mobile devices with limited memory, a modest growth rate option is used. HR can cause damage to certain eye areas. If these damaged areas are not recognized at an early stage, hypertensive retinopathy develops. Figure 1 displays images of a clean retinal fundus and images showing symptoms of an eye disease brought on by HR. 

Ophthalmologists can find different retinal diseases, such as those linked to HR, with the help of computerized methods [8]. These technologies aid academics and the global medical profession by enabling self-diagnosis. Optometrists use these technologies to treat and diagnose eye-related illnesses, particularly those that are HR-related. Hypertensive retinopathy (HR) can be identified by segmenting the retina’s structural features, such as the macular, optic nerves, arteries, and vasculature, as illustrated in Figure 1. There are two ways of extracting features from eye images: using either deep-learning models or handcrafted feature extraction techniques. In the case of using deep-learning techniques for feature extraction, these features can be statistically evaluated to determine the best features to identify HR or non-HR illness. Deep learning techniques have been applied for many applications, such as computer vision algorithms [9], biological behavior analysis, and many other different applications. A visual diagram of (**a**) normal and (**b**) hypertensive retinopathy (HR) is displayed in Figure 2.

### 1.1. Clinical Implications of Hypertensive Retinopathy

To accurately identify organ damage caused by hypertension before symptoms appear, experts utilize different techniques in clinical practice. The only organ in the body where systemic hypertension-related vascular alterations can be seen in action is the eye. The key factors affected by systematic hypertension are discussed in Table 1. In addition, the severity level of HR has also been adopted in the past, which is used in clinical practices. These grading schemes are briefly described below: (1)Grade 1: mild widespread constriction of the retinal arteries.(2)Grade 2: arteriovenous pinching and definite focal constriction.(3)Grade 3: retinal hemorrhages, exudates, and cotton wool patches, in addition to grade 2 retinopathy symptoms.(4)Grade 4: severe retinopathy of grade 3 with papilledema.

However, in this paper, we focused on the recognition of hypertension retinopathy (HR) instead of grades of HR. In fact, it is very difficult to create a balance dataset of HR in terms of severity grades. The stages of HR have not commonly been addressed in the past. Clinically, the vasoconstrictive stage is characterized by widespread vasoconstriction and elevated retinal arteriolar tone, which were acutely induced by autoregulatory mechanisms. After, continued high blood pressure results in sclerotic alterations, such as intimal thickening and hyaline degeneration, which then leads to more arteriovenous nicking or nipping. The blood–retinal barrier is broken during the exudative stage of the process, which results in the production of blood and lipid exudates that cause retinal ischemia. Compared to HR, diabetic retinopathy (DR) is also a very complicated eye-related disease. In fact, clinical experts use handcrafted techniques to differentiate between HR and DR eye-related diseases when diagnosing using retinographic images. However, this is still a time-consuming task. Therefore, computer-aided diagnosis (CAD) systems were developed to address this issue. In this paper, we developed an automatic CAD system to recognize HR eye-related disease.

### 1.2. Background

Several methods have been discussed in the past that use pretrained architecture with a transfer learning (TL) scheme to automatically classify HR eye diseases when diagnosed with retinography. This fits with the general trend of making networks deeper and more complicated to improve the accuracy. For the categorization of HR, several publications used pretrained CNN models or customized CNNs. IoT and embedded devices are currently used extensively. Unfortunately, their CPU and storage capacities are typically inadequate. These devices cannot use more complex networks due to the number of parameters and processing demands. It is conceivable to examine special network architectures that offer maximum accuracy with very tight computational cost constraints to satisfy the application requirements. Compact and efficient neural networks, such as SqueezeNet, MobileNet, and ShuffleNet [10], were created to overcome these issues. These techniques offer a small architectural unit that may be added to current networks to boost performance at a small expense. In this study, we propose a new architecture that combines the DenseBlocks method with efficient MobileNet CNN networks that have already been trained (in terms of latency and memory size). We chose MobileNet as the backbone because it has efficient topologies that make networks small and allow for more data to be encoded. Trials comparing other HR classification schemes have shown that the proposed lightweight Mobile-HR model performs better than those schemes. This result shows that the accuracy can be improved. Lastly, a real-time application demand may be satisfied by Mobile-HR.

### 1.3. Major Contribution

In this research, the Mobile-HR system was designed to solve the problems listed above. It does this by sorting data into HR and non-HR using MobileNet architecture and dense blocks instead of focusing on image processing methods. Below are some of the Mobile-HR system’s most significant contributions.

In this study, the authors gathered a huge dataset from Pakistani hospitals (named PAK-HR) and internet sources. With the help of the 9170 photos in this dataset, the trained model was able to be very accurate.In this study, the MobileNet architecture was made by putting together dense blocks to make the Mobile-HR system’s multilayer architecture. The complex architecture of the Mobile-HR model was changed to find HR-related eye disorders by adding dense blocks.The method used in this work to classify HR is based on deep characteristics and a color space that is geared toward how people see things. As far as we know, this is the first time anyone has tried to make an automated system for identifying HR diseases that works better than other methods described in the literature.Before it is used, Mobile-HR is trained with a huge number of HR retina images. This makes the model more generalized compared to state-of-the-art approaches.Mobile-HR has a very high level of accuracy (99%), which is higher than any other method that has been suggested in the literature.

### 1.4. Paper Organization

The rest of the sections of this paper are organized as follows: Section 2 describes the literature survey of articles related to this research; Section 3 introduces the proposed architecture; Section 4 shows the experimental results and compares our work with state-of-the-art research; Section 5 discusses the results; and Section 6 is the conclusion.

## 2. Related Work

DL-based image diagnosis methods were developed in the past to help doctors make more accurate HR diagnoses. Before 2016, most studies used standard methods for preprocessing, segmentation, feature extraction, and classification. Public access to various skin lesion databases is now possible. To distinguish between HRs, researchers have developed DL algorithms [11]. Over time, it became clear that the CNNs retrieved more useful characteristics than the handcrafted techniques. The DL- and TL-based approaches for diagnosing HR were recently used in the research we looked at and chose for this article (Table 1). The paragraphs that follow provide a brief description of the investigations.

Many research articles [12,13,14,15,16,17,18,19,20,21,22,23,24,25,26,27,28,29,30,31,32] have focused on how to make small, effective networks that can be used for many different things. Researchers have tried out a wide range of methods, such as training network models and making older models smaller. Andrew et al. [13] came up with an efficient way to design lightweight neural networks that can be used in a number of computer vision applications, such as fine-grained classification and object identification, to achieve the benefits of small, low-latency models. Some authors have used the fuzzy inference method to determine how likely it is that diabetes will cause serious problems.

After looking at data with the VGG-19, MobileNet, and Resnet models [15], Wu and Hu [14] came up with a transfer learning classification method for hypertensive retinopathy. When the transfer learning approach is applied to the Kaggle dataset, the experimental accuracy is 60%, which is better than the model’s initial learning. Sun and Zhang published a model to detect hypertensive retinopathy [16]. Five distinct algorithms, including decision tree, random forest, support vector machine, logistic regression, and naive Bayesian, were utilized in electronic health records from 201 institutions [17] to improve diagnosis. They compared these models, and out of the five models, the random forest model had the highest accuracy (92%). In using the retinal vasculature as a crucial indicator for intelligent, DL-based screening and analysis of the diagnosis of diabetic and hypertensive retinopathy [18], when compared to cutting-edge techniques for automated vessel detection for diagnostic purposes, the authors’ accuracy results demonstrated the suggested method’s remarkable segmentation capability.

Mukesh et al. [19] proposed a system to detect HR lesions by suggesting a regional IoT-enabled federated learning-based categorization strategy (IoT-FHR) that integrates both global and local features. To improve the effectiveness of the classification of IoT-FHR, the local feature arterial and venous nicking (AVN) classification model was fused with the general IoT-FHR classification model. When evaluated on a private dataset, the recommended fusion model’s accuracy, sensitivity, and specificity were 93.50%, 69.83%, and 98.33%, respectively. Joseph et al. [20] proposed a machine-learning-based automated method for HR detection using fundus photographs. The study emphasizes how an early diagnosis of some medical issues using only a photograph acquired from the fundus image of the eye is more effective when conducted using computer-automated methods rather than manual observation techniques. In their study, Arslan et al. [21] suggest the dual-stream fusion network (DSF-Net) and the dual-stream aggregation network (DSA-Net) as two new shallow-DL architectures that can be used to recognize retinal vasculature. Semantic segmentation is used to find diabetic and hypertensive retinopathies in raw color fundus images. The effectiveness of the suggested strategy was evaluated using three publicly accessible metrics. The results of the experiments further demonstrate that the DSA-Net offers greater SE in comparison to the current methods. This study developed a system that is not as deep as standard semantic segmentation networks but delivers acceptable segmentation with limited trainable parameters and layers. Compared to existing methods, the proposed approach outperformed them on three publicly accessible datasets in terms of sensitivity, specificity, area under the curve, and accuracy metrics.

The study in [22] stated that deep learning ideas could be used to find HR and showed that the validation sensitivity was 95%. The most important thing that this study adds is the preprocessing step of adaptive histogram equalization. Some other interesting work in HR detection using machine learning models can be found in [23,24,25,26,27]. Qureshi et al. [28] developed a way to find HR using fundus images based on a depthwise separable CNN network. Their work reported a 95% accuracy and a 0.96 AUC. In [29], the authors propose using a fundus image as input for a CNN to classify images into HR and non-HR images. The proposed system was tested using the DRIVE dataset and based on experiments; the accuracy of the proposed model reached 98.6%.

Abbas et al. [30] came up with a new way to find hypertensive retinopathy (called DenseHyper) in retinal fundus images. Through ten-fold cross-validation, the proposed work performed much better than other algorithms, with an average accuracy of 95%. Recently, Arsalan et al. [31] came up with a new way for computers to help diagnose diabetes and high blood pressure retinopathy. They used shallow neural networks that are easy on memory, pool-less segmentation networks (PLS-net), and pool-less residual segmentation networks (PLRS-net). The DRIVE, CHASE-DB1, and STARE databases, which are all publicly accessible, were used for the research. The PLRS-net outperformed PLS-net by averaging an 82% sensitivity across all three datasets, which was better than PLS-net’s performance. The authors of [32] also created a five-step HR recognition system using semantic and instance segmentation in the DenseNet architecture. 

There are three ways to classify segmented images in the literature [33]: Efficient-Net, VGG-16, and ResNet-152. Using the ensemble method, the generated feature vectors were combined, and the SoftMax classifier was used to accurately classify eleven different types of retinal disorders. The suggested method for recognizing HR had an accuracy of 99.71%, a precision of 98.63%, a recall of 98.25%, and an F measure of 99.22%. In [34], on the other hand, the authors used a segmentation method for blood vessels without considering any other HR properties. 

## 3. Materials and Methods

In this paper, the Mobile-HR system is suggested as a way to classify eye problems such as hypertensive retinopathy. The Mobile-HR was built by combining both MobileNet architecture and dense blocks, as shown in Figure 3. In the proposed Mobile-HR system, the effective features are extracted using deep learning (DL) techniques. Transformational learning is used here, as we train an already trained model using the proposed new dataset PAK-HR. The Mobile-HR architecture consists of seven key stages built to identify related HR features from retinal fundus images. Integrating the characteristics acquired from both the MobileNet architecture and dense blocks using the component multiplication method are the main contributions of the proposed Mobile-HR system. The dimensions of the dense blocks are continuously changed throughout the training. The final step is to add the SVM classifier layer with a liner activation function to classify the image as HR or normal. The linear activation function is added to smooth the procedure and improve the classification results. 

### 3.1. Data Acquisition

A 9170 retinal fundus image dataset (3410 HR images and 5760 non-HR photos) from a variety of reputable hospitals in Pakistan and from well-known internet sources was acquired to train and assess the performance of the Mobile-HR model. The training dataset was created with the help of a professional ophthalmologist (manual separation of HR and non-HR fundus images from several datasets). In Table 2, we can see the breakdown of the three datasets (with different dimension settings) that were utilized to construct our testing and training fundus set. After processing, binary labels were given to these pictures. Data augmentation was proposed here for balancing the total number of images with and without the disease, and this was conducted to ensure that the dataset was objective. Images from the dataset were reduced in size to 700 × 600 pixels for preprocessing before being delivered to an algorithm designed especially for the Mobile-HR model. By experimental analysis, we determined that the perfect image downsize was to 700 × 600 pixels. It is frequently wiser to reduce the size of larger photos to match that of tiny images rather than making small images larger. In practice, the DL models generally train more quickly on tiny images.

As mentioned earlier, data from Pakistani hospitals were used to build the total dataset used to train and evaluate the proposed Mobile-HR. These images were taken as part of routine testing for hypertension. These data included 5590 retinal samples; 2100 were from HR patients and the remaining 3490 were from non-HR patients. All of the data are JPEG files that were saved at a resolution of 1125 × 1264. Furthermore, data on Imam-HR were used to train and evaluate the proposed Mobile-HR system. The dataset includes 3170 retinal samples; 1130 were from HR patients and the remaining 2040 were from non-HR patients. All of the data are JPEG files that were also saved at a resolution of 1125 × 1264. Using pictures from these different sources, the complete PAK-HR was introduced. Figure 4 shows the HR fundus image used for the proposed model. 

### 3.2. Preprocessing and Augmentation

In this step, the raw data were removed from the fundus images. In addition, the images were cleaned using the flip-flop method and filters for further processing. Moreover, this step involved setting the missing or incorrect values of the pixels and removing the outliers. It also included feature engineering, such as the normalization of variables and the selection/creation of new features that can improve the accuracy of the algorithms. Figure 5 illustrates the preprocessing steps along with the data augmentation techniques. 

In preprocessing, different operations are applied to images, such as cropping, contrast, horizontal flip, spin, pan, and enhance using filters. Cropping involves removing unwanted parts of the image so that only the desired sections remain. Contrast acts to adjust the brightness levels in the image, while vertical and horizontal flips swap the orientation of the image along with its axes. Panning is a technique that zooms in or out from an area of focus, while embossing adds depth and texture to an image by shifting pixels up or down. All these processes helped to improve the quality of the images and increase the classification accuracy. The parameters are shown in Table 3.

### 3.3. Mobile Net and Dense Block

MobileNet is a convolutional neural network architecture that uses depthwise separable convolution as the foundational building block. Convolution that is depthwise separable contains two layers: point convolution and depthwise convolution. As shown in Figure 6, the Mobile-HR model treats two distinct convolution layers: depthwise convolution layer and point convolution layer. Each depthwise convolution layer in the dense block uses the output feature maps from the preceding layer as its input feature maps. Although depthwise convolution only uses one channel, the sum of all the output feature maps from the layers preceding it is the number of input feature maps for the middle depthwise convolution layer, which is equivalent to the number of input feature maps. The Mobile-HR model employs four convolutions as a dense block and decomposes a depthwise separable convolution into two distinct layers. Figure 7 displays the full architecture of the proposed model (Mobile-HR).

### 3.4. Mobile-HR Architecture

All steps in the form of Algorithm 1 are presented in the proposed Mobile-HR model to extract the deep features map. The real output value is displayed by the function O(y), and the dense layer learning is denoted by R(y) in network input y. Figure 8 represents the dense block used in the proposed work. In this model, we added the 3Conv2D, average pooling 2D layer, and flattened the Dense and Dense_1 layers after the pointwise 13 convolution layer, which shows better feature results. A visual representation of the dense block is shown in Figure 9. The features are selected from this model and then the model evaluation SVM classifier is employed on this. Using a training–test splitting method of 75% to 25%, the linear SVM machine learning classifier was utilized for the automatic classification of HR. Because of its high performance and ability to handle small datasets, linear SVM is frequently employed.
(1)O(y)=R(y)+y


**Algorithm 1: Implementation of the proposed Mobile-HR model for feature map extraction**

**Output**
Feature map extraction y = (y1, y2 …, yn)Step 1Input normalization of raw dataStep 2Function definitionStep 3Kernel sizes and array Y, which comprise several filters, are the inputs to the conv-batch norma. Y = Conv (Y) andb. Y = BN (Y) are then appliedStep 4Depthwise Conv2D was used rather than Conv2DStep 5Establishing the networka. 14 Convolution layers, each comprising 32,64128,256512,1024 filters, make up the first step of the procedure. After each of them, the ReLU is subsequently activated.b. The next step is to use Add to use skip connectionc. Three distinct skip connections are utilized. Each skip connection has three depthwise convolution layers after the Maxpool layer. The skip connection has two strides and a conversion ratio of 1 to 1.Step 6After, the flattened layer is used, the feature map and Y = (y1, y2 …, yn) are created and flattened at the end[End Feature map extraction function]

Support vector machine (SVM) is a machine learning classification strategy that outperforms other types of classifiers and is commonly used to tackle real-world issues. In this paper, we used SVM to recognize features into hypertensive retinopathy (HR). All of the steps are presented in Algorithm 2. For computer vision or image classification challenges, we developed a depthwise separable CNN instead of a deep learning or machine learning classifier. Given that we were working with binary classification issues, it seemed logical to use linear SVM. Another reason for employing linear SVM was to improve the efficacy of our approach and to identify the optimum hyperplane that divides the feature space of ill and normal cells in retinal pictures. An SVM generally accepts a vector T = (a1, a2…, an) and produces a value, t c Rn, which may be written as:(2)Tout=(Weig,Aiv)+c

The *Weig* parameter represents the weight, and c represents the offset in Equation (2); both the *Weig* and *c* parameters belong to R and are learned during training. The *Aiv* parameter is the input vector, and it is allocated to class 1 or class 1 depending on whether y is larger than or less than 0. To create the best data separation hyperplane, we must minimize:(3)$$Ueig=\frac{1}{2}{\left||Weig|\right|}^{2}$$

The mechanism of our proposed classifier is described in depth in Algorithm 2.
**Algorithm 2: SVM Classifier to Recognize hypertensive retinopathy of the extracted features****Input***Extracted feature map x = (a_1_, a_2_,. →, an) with annotations a = 0,1. Test data A test*Output*Recognition of hypertensive retinopathy (HR) and normal retinographic samples*Step 1*Primarily, the SVM classifier and Kernel Regularize L2 parameters are defined for optimization*Step 2*Classification of normal and abnormal samples*Step 3*Depthwise Conv2D was used rather than Conv2D*Step 4*Building classifier based on SVM**a. The training process of SVM is completed using extracted features**t = (a1, a2,.., an) by our Algorithm 1. b. For the generation of the hyperplane, use Equation (6).*Step 6*The class label is allocated for testing the samples with z-test**using the decision function of the equation below:**A* *test**= (Weig, Aiv) + c*

## 4. Experimental Results

A 9170-image dataset, obtained from a variety of reputable hospitals in Pakistan and from well-known online resources, was used to train and test the Mobile-HR model. Binary labels were given to these images after processing by an expert in the field of HR classification. The dataset is composed of a total of 9170 images, of which 3410 were used to test the system. The images from the dataset were compressed to 700 × 600 pixels for preprocessing, and then they were sent to the preprocessing algorithm created specifically for the Mobile-HR model. To reduce the variability across data points, the images were normalized.

A dataset of 3410 retinal pictures, both HR and non-HR (as shown in Figure 10), was used to train the Mobile-HR. These retinography photos were acquired from several reputable hospitals in Pakistan (Pak-HR), as well as from reputable online resources. All 9170 images were reduced to 700 × 600 pixels to execute the feature extraction and categorization activities. The Mobile-HR system was built by combining MobileNet and dense blocks. The Mobile-HR model was trained for 100 epochs, with the best model being discovered in the 20th epoch and having an f1-score of 0.99. To assess the effectiveness of the proposed Mobile-HR system, the accuracy (ACC), specificity (SP), and sensitivity (SE) ratings were computed using statistical analysis. The created Mobile-HR system’s performance was measured against these metrics and compared to that of other systems. A computer with an HP-i7 processor, 8 cores, 16 GB of RAM, and a 2 GB NIVIDA GPU was utilized to construct and develop the Incept-HR system. Windows 11 Professional 64 bit was installed on this machine.

For the MobileNet-HR transfer learning model, Figure 11 shows the accuracy with loss of the suggested training and testing. The experiment was carried out with the use of the 10-fold cross-validation technique. The training accuracy (Acc), testing accuracy (Val Acc), training loss (Loss), and testing loss were the measurements for each fold, encompassing the training and testing samples (Val Loss).

The 10-fold cross-validation test was used in the studies to divide the data into two groups for the training and test sets. The test set was used to assess the model and make predictions. In addition, we evaluated the classification accuracy following the input cropping of the core region of interest (ROI) to 700 × 600 pixels. The hyperparameters were standardized across all networks. The network models were trained using stochastic gradient descent (SGD), which runs rapidly and converges well. We trained the networks in 64-batch increments because of GPU memory limitations. The learning policy for all networks was “step” with a gamma of 0.5, and the starting learning rate was set to 0.001. Several optimization setups were also applied. Using the ADAM optimizer on a categorical cross-entropy loss, the networks became more efficient. We used a weight decay of 5 × 10^−4^ and a Nesterov momentum of 0.9 to perform the comparisons. The BN method, ReLU, and GELU functions were used in every experiment.

We evaluated the performance of the recommended Mobile-HR classifier against cutting-edge alternatives using a variety of statistical criteria. Several measures have been employed in the past, including accuracy (ACC), recall, specificity, precision, and F1-score. These measures were used in this study to compare with cutting-edge systems. The true positive (TP) and true negative (TN) values, which show whether the model was effective in predicting whether the data was genuine or false, were used to produce these metrics. The FP and FN signs demonstrate that the system incorrectly forecasted the data, whether it was true or false. To put it another way, it is a method for determining how well the algorithm classifies the data. Moreover, enhancing the model quality lowers the possible high cost of errors. These statistical indicators are calculated in the following way:(4)Accuracy=(TP+TN)/(TP+TN+FP+FN)×100
(5)Recall=TP/(TP+FN)×100
(6)Specificity=TN/(TN+FP)×100
(7)F1−Score=2×(precision×recall)/(precision+recall)

### 4.1. Experiment 1

In the first experiment, we report the performance of our proposed model using 10-fold cross-validation. The area under the curve (AUC) was the primary metric used to evaluate the classification performance. Table 4 displays the results of our quantifiable evaluation of the produced Mobile-HR system’s performance. The developed Mobile-HR model had a low training error (0.1) and high AUC (99%) for detecting HR eye disease.

### 4.2. Experiment 2

In this experiment, we trained two deep learning models (VGG16 and VGG19) and assessed their performance against the proposed Mobile-HR system. The fact that these deep learning models were trained using the same number of epochs is noteworthy. Table 5 displays the findings of a comparison of the VGG16 and VGG19 models with the Mobile-HR system in terms of the sensitivity, specificity, accuracy, and area under the curve (AUC). Figure 12 shows the training validation loss and accuracy for VGG16 and VGG19, respectively. The comparison demonstrates the Mobile-HR’s edge over VGG16 and VGG19 in terms of performance.

### 4.3. Experiment 3

We began testing our proposed Mobile-HR model on the DRIVE and DiaRetDB0 datasets using the training and validation accuracy, as well as the training and validation loss functions. As seen in Figure 13, our proposed model performed well, achieving a training and validation accuracy of 100% while requiring just ten epochs. In addition, the proposed model was able to reach a loss function for both the training and validation data that was less than 0.1, demonstrating the effectiveness of our suggested model.

### 4.4. Experiment 4

We used the Imam-HR dataset, a separate one, for this experiment. We initially investigated the accuracy of the model. 

Figure 14 depicts the training, validation accuracy, and confusion matrix of the Mobile-HR model using the Imam-HR dataset. We obtained a 100% accuracy on the training and validation data, suggesting that our approach works effectively on pictures from the Imam-HR retina dataset.

### 4.5. Final Experiment

In this experiment, we assessed the efficiency of our suggested Mobile-HR system using a new dataset called Pak-HR that was gathered from Pakistani hospitals. Using both sets of data, we first compared the model’s performance on the training and validation sets, as well as the loss function. Figure 15 displays the training validation accuracy, loss, and confusion matrix of the Mobile-HR model. We were successful in achieving flawless accuracy on both the training and validation sets using the retina dataset.

### 4.6. State-of-the-Art Comparisons

The research on applying deep learning methods to find HR-related features in retinal images is a new research direction. Triwijoyo-2017 [29] and CAD-HR [30] are the state-of-the-art works related to this research area. The most recent deep learning model for HR detection is termed CAD-HR. It can be seen from Table 6 that Mobile-HR had a superior performance over CAD-HR and Triwijoyo-2017 [29].

In comparison, the developed Mobile-HR system achieved excellent results, with corresponding SE, SP, ACC, and AUC scores of 99%, 99%, 0.99, and 0.99. According to [29], the identification accuracy of CAD-HR was 95%. In [30], the authors employed a very restricted collection of input images for training, as they used only 40 retina images that were divided into 20 that were normal and 20 that were HR. This small number of dataset images led to the small precision and accuracy reported by the researchers. In addition, this dataset was not approved or classified by an expert optometrist. Therefore, with the approval of expert optometrists, our Mobile-HR system was tested and trained on a balanced 9170-image dataset. As a result, we achieved a 99% accuracy in our classification, which is considered a large improvement over state-of-the-art works.

In another test, we compared how the CPU, GPU, and TPU handled the proposed Mobile-HR model in terms of batch size and processing speed. In practice, the CNN implementation in the CPU, TPU, and GPU had to be looked at layer by layer. The Mobile-HR network should be built with each job being a multiple instruction, single-data (MISD) task. Prioritizing the neural network’s tasks is necessary while building the network. The GPU provides greater flexibility and straightforward programming for modest numbers. GPUs better fit batch sizes for fewer data because of the execution pattern in the wraps and the scheduling on straightforward on-stream multiprocessors. The GPU performs well for large datasets and network models by optimizing the memory reuse. Fully connected neural networks have a lower weight reuse, which causes increasing memory traffic as the model size increases. The GPU may be utilized for applications requiring memory because of its bandwidth. For processing large neural networks, GPUs outperform CPUs due to the added parallelism capacity. With fully connected neural networks, the GPU performs better than the CPU, while the TPU shines when dealing with large batch sizes. On the other hand, we used an array structure for the TPU because it works better with large batches on the Mobile-HR architecture and provides a high throughput during training. For the matrix and multiply units in the TPU’s systolic array to work well, they need to be given a lot of data at once. When the batch size increases, the architecture speeds up. Due to the networks’ ability to reuse space for big batch sizes and intricate CNNs, TPU is the best. In Table 7, the proposed Mobile-HR model’s performance is recorded.

We suggested deep blocks, which were tested on a large HR dataset for the classification of HR eye-related disease to produce cutting-edge results. For example, when SE modules were added to the main CNN and LSTM models, Mobile-HR’s top-1 error rate went up to 8.7%. Despite a minor reduction in the theoretical complexity, we found that many DL models, as detailed in Table 8, are frequently 25–40% less efficient than the suggested model on mobile devices. This shows that the real-time speedup assessment is essential for developing inexpensive architecture.

The decision-making process used by Mobile-HR is evident from the heat map of the hypertensive retinographics and the normal picture (as shown in Figure 16). Nonetheless, from Figure 16 it is evident that the suggested Mobile-HR system can operate well with HR retinograph pictures, which is advantageous for a crucial biological application such as this one. On the other hand, because they make it simpler to classify disorders using retinal fundus pictures, HR image classification is superior for computer-aided diagnosis (CAD). An expert ophthalmologist confirmed the heat maps generated from the proposed Mobile-HR model to distinguish between the normal and HR retina fundus images.

## 5. Discussion

Our knowledge of the epidemiology, systemic linkages, and clinical consequences of hypertension eye illness, particularly hypertensive retinopathy, has significantly improved as a result of developments in research over the past three decades. Hypertensive retinopathy, which is traditionally diagnosed by a clinical funduscopic examination but is increasingly being recorded on digital retinal fundus pictures, has long been thought to be a sign of systemic target organ damage (such as kidney disease) elsewhere in the body. According to epidemiological research, hypertensive retinopathy symptoms are often observed in the general adult population, linked to subclinical vascular disease indicators, and foretell the likelihood of incident clinical cardiovascular events. With the advancement of noninvasive optical coherence tomography angiography, artificial intelligence, and portable ocular imaging devices, the ocular manifestations of hypertension have been better assessed and understood, increasing the possibility that ocular imaging will be used to manage hypertension and categorize cardiovascular risk.

Consequently, we developed an improved CAD system to diagnosis HR from retinal fundus images to address many limitations of the past systems. Several limitations of state-of-the-art studies on hypertensive retinopathy are described in Table 9. To overcome these limitations, we developed a novel Mobile-HR system. The Mobile-HR system uses a trained CNN model called MobileNet to classify HR images. The model was built using four successive dense blocks and an output layer that was fully linked. Without the aid of a specialist, this multilayer architecture automatically uses learning processes to extract features from the input picture. The remaining blocks were added to the original model to increase more universal outcomes and features for Mobile-HR architecture. Convolutional, pooling, and fully connected layers make up the majority of the CNN model, which is utilized to learn deep features. These layers must be trained and proven to be successful at extracting useful characteristics before being used to build the model. An independent feature learning technique allowed for success in detecting HR. This makes our approach superior to handcrafted-based classification systems that depend on the preprocessing, segmentation, and localization of HR-related data, which are time-consuming and complex techniques for diagnosing HR disease. There were a number of significant issues when HR automated systems were developed using conventional methods. The first issue is that, even with the use of sophisticated pre- or post-image processing methods, it is incredibly difficult to recognize and extract important HR-related lesion features from retinographics. The second issue is that there are no datasets with clinical expert labeling to explain specific HR-related lesion patterns; therefore, the network cannot be trained or tested. Our system solves both issues; first, we propose using deep learning models for the extraction of important features from the eye. Second, we introduce a new dataset: PAK-HR. Several models are introduced in the literature to learn feature extraction. All previous models use the same weighting technique at each stage. This makes it challenging for layers to transmit weights to the deeper network layer. This study develops the Mobile-HR system, which uses two multilayer deep learning approaches to distinguish between HR and non-HR without concentrating on techniques for image processing to overcome the problems. There are several significant contributions proposed in the Mobile-HR system. To the best of our knowledge, the Mobile-HR system is the first attempt to classify HR data using Mobile networks and dense blocks combined. This proposed new model acquires four distinct HR-related injuries as features for the system. The model builds a feature map, establishes the precedence of features, and enhances the learning process’s effectiveness. In this research, the proposed model must be trained on many samples before it can be used to build the Mobile-HR system. This is so that the learned features will be more general. Future work on the proposed Mobile-HR system might include adding a greater selection of retinography images that have been obtained from diverse sources. In addition to deep characteristics, the model may also include handcrafted features to improve the classification performance. In future work, we may add the saliency maps approach [29,30] to extract HR-related features. This method will be added to the system to improve how well HR eye-related sickness is categorized. In addition, the classification of the severity of HR sickness will be considered in the future. Several investigations have shown that clinical features are important factors in influencing the level of HR intensity. Yet, it is difficult to remove such HR-related lesions with different thresholds to gauge the severity of HR sickness.

### 5.1. Advantages of the Proposed Approach

To obtain accurate results from the classification of HR, data imbalance is a key step. Deep learning (DL) methods use several artificial neuron layers to recognize pictures effectively. However, by using the data augmentation approach to (700 × 600, 3) pixels, the class imbalance has been corrected. If datasets are not used for this stage, then additional processing power and memory are required. Due to the fact of their complex architecture, deep learning (DL) and machine learning (ML) algorithms frequently overfit. Our responsibility is to offer a less complex structure and speed up calculations. We thus propose our Mobile-HR design with a balanced layer architecture. To increase the speed, we added a variety of blocks with kernel regularizations of 0.001 to the design. Kernel regularization’s primary goal is to solve overfitting issues. We used an SVM classifier in the study to identify HR due to the noise in the retinograph images. Consequently, all sorts of optimizers work equally well for us, except for GELU, and our model uses it to provide results quickly. The study makes use of three well-known datasets that are available to the public. The recommended deep Mobile-HR DL model performs better on these datasets than competing models. The following benefits of the suggested study are briefly explained below:1.The most important part of our work (HR) is the idea of a new, highly optimized, and lightweight CNN model that can recognize hypertensive retinopathy. Compared to other deep learning (DL) architectures, the Mobile-HR design makes networks less complicated while improving their accuracy and speed through dense blocks. The dense block of our model’s Mobile-HR architecture modifies and increases the accuracy. Moreover, it has little effect on the model’s complexity or recognition rate.2.Mobile-HR has a generic capability with no overfitting or underfitting issues. Since the activation function makes the model more accurate, we changed the original ReLU function in the suggested model to the GELU function instead, which, according to our study, made the model better. When the activation function is there, the model better recognizes the HR class.

### 5.2. Limitations of Proposed Approach and Future Works

There are several potential types of HR. This study only includes only distinct between HR and non-HR eye-related disease instead of varieties of eye-related disease. The performance of the suggested TL-based model will be evaluated in the future using the additional classes of HR. Furthermore, it is generally known that any DL-based approach requires a large amount of data to train the model successfully. On the other hand, the study did not use enough photos to train the recommended model. In this work, we added a dense blocks mechanism to the MobileNet model, which improved its accuracy compared to other recognition models. Although we did well in terms of the complexity of the model and the speed of identification, the accuracy may still need improvement. One of the things we learned was that the dataset collection size we used was still too small. Additionally, our method does not evaluate the five classes of HR. Addressing this issue, several datasets and academic publications are not enough to discuss the identification of various HR classes. All datasets are susceptible to the issue of an unbalanced sample distribution. To counterbalance it, we used a data augmentation strategy. However, another balanced dataset is required. Nothing can truly be done with the dataset at hand to alter this circumstance. 

Future research may focus on analyzing our proposed model using larger, more representative classes of HR. Regarding the second issue, it should be highlighted that we considered that improving the analysis of timing data will undoubtedly increase the amount of computation required for the model, undercutting our objective for it to be lightweight while optimizing MobileNet. Increasing the dataset’s size and using lightweight models for analyzing HRs data are two further potential study areas. Since Mobile-HR is a thin neural network, future research based on compact devices—such as tablets and portable devices with GPU processors—might be considered to ascertain if it is feasible to create small engagement detection devices. In addition, this work suggests a Mobile-HR model based on a pretrained method, which has recently gained popularity. The development model is computationally efficient and successful for the deep feature classification of multiclass HR lesions. To assess this model’s accuracy and computing efficiency; however, it is important to do so with a graph-based method. The next steps will handle this action. 

In addition, there were fewer individuals with alleviated HR than there were patients with chronic hypertension, despite clinical data showing a strong correlation between the grades of HR with the prognosis of hypertensive patients. We cannot completely rule out the potential that retinal alterations occurred before a diagnosis of HR because the research was retrospective in nature. Although we removed individuals with low-grade HR, it is possible that we overlooked minute retinal vascular alterations. Some of the individuals could have had prior hypertension episodes that led to retinal thinning. Therefore, retinal ischemia or previous hypertensive episodes may have caused the retinal thinning. Once more, we cannot completely exclude earlier hypertensive episodes. Fluorescein angiography, which can detect retinal ischemia, was not conducted. If there was a connection between ischemia and retinal thinning, it would have been more obvious if we had performed that. Additionally, we did not carry out long-term follow-up on the prognosis of the hypertensive patients, thus we were unable to assess if the alterations were prognostic or to monitor the proper course of therapy. We also skipped additional elements that have been demonstrated to cause ischemia damage to the retina, such as smoking and hyperlipidemia.

## 6. Conclusions

In the literature, few fully automated systems to identify HR from colored images have been proposed. Most of the work proposed in the literature concentrates on obtaining features (such as arteriolar-to-venular diameter ratio, arteries, optic nerves, cotton wool spots, microaneurysms, vascularity, and hemorrhages) from images and employing some machine learning algorithms to classify the image depending on these features Because of this, the identification system for hypotension will be constructed using feature selection and image processing expertise. Few systems use deep learning models to extract features from images such as the proposed system. Adding deep learning as a screening tool for HR detection is challenging, since they need a large dataset to train to reach high accuracy. In this work, an innovative computerized method for HR (Mobile-HR) has been created to address these issues. The suggested system uses dense blocks with MobileNet architecture to classify images. Even though network model acceleration and compression decrease classification accuracy, adding dense blocks to the model provides improved performance. The results show the superior performance of the proposed system over state-of-the-art proposed systems. The system reaches 99% on the proposed challenging dataset.

## Figures and Tables

**Figure 1 diagnostics-13-01439-f001:**
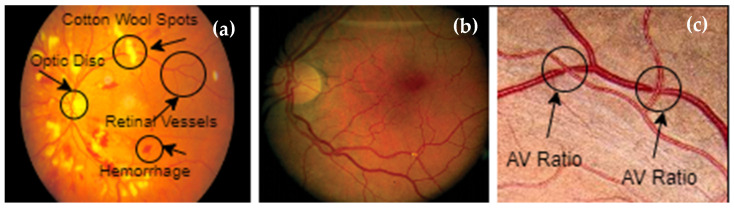
Illustration of the vascular system: (**a**) optic disk, cotton wool spots, and hemorrhages; (**b**) tortuosity; (**c**) A/V ratio.

**Figure 2 diagnostics-13-01439-f002:**
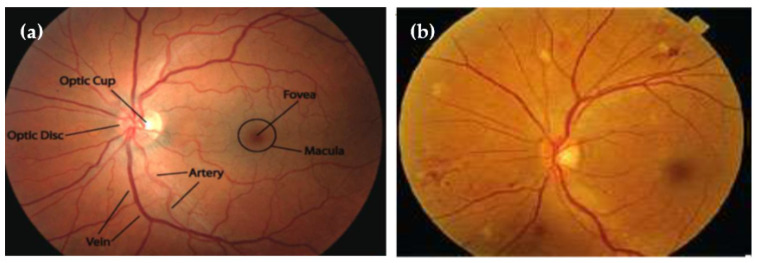
A visual diagram of (**a**) normal and (**b**) hypertensive retinopathy via diagnosis by retinal fundus images.

**Figure 3 diagnostics-13-01439-f003:**
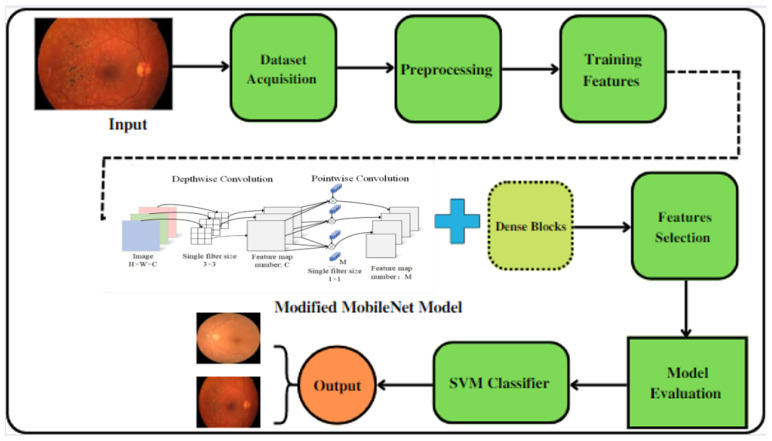
A systematic flow diagram of the proposed Mobile-HR system for the diagnosis of hypertensive retinopathy eye-related disorders.

**Figure 4 diagnostics-13-01439-f004:**
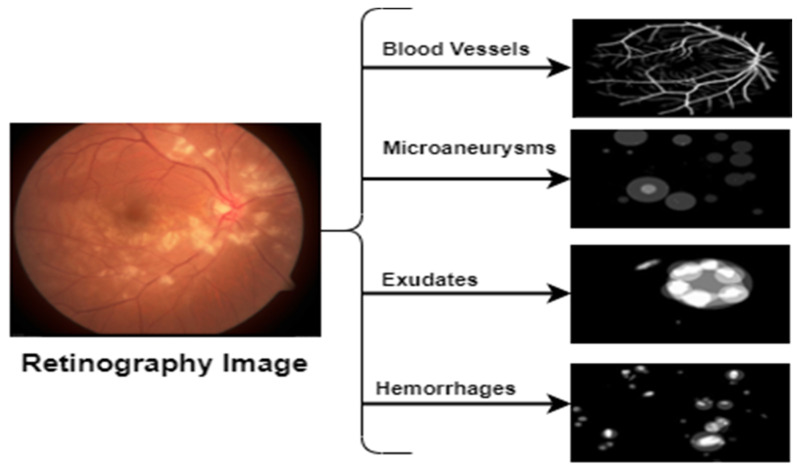
An HR fundus image used for the Mobile-HR system’s training.

**Figure 5 diagnostics-13-01439-f005:**
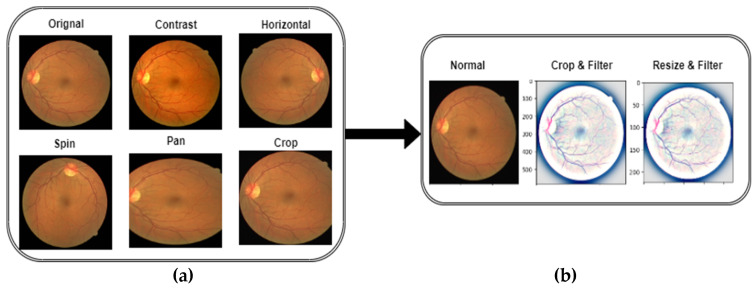
A visual diagram of the preprocessing steps to apply the (**a**) data augmentation techniques and (**b**) enhance the HR-related lesions.

**Figure 6 diagnostics-13-01439-f006:**
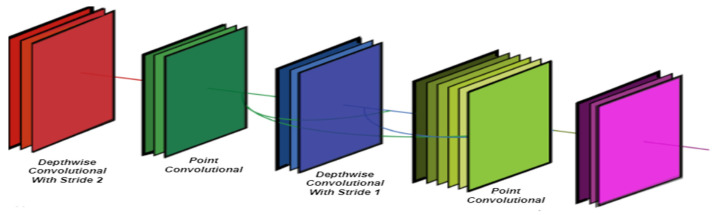
A visual diagram of the architecture of MobileNet.

**Figure 7 diagnostics-13-01439-f007:**
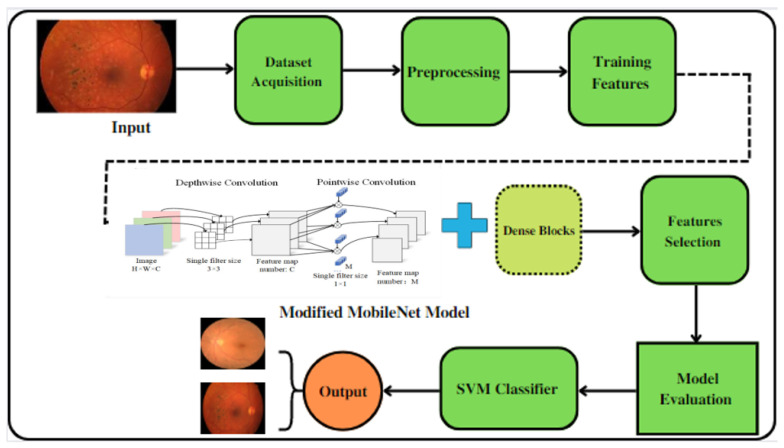
A schematic flow diagram of the proposed Mobile-HR system.

**Figure 8 diagnostics-13-01439-f008:**
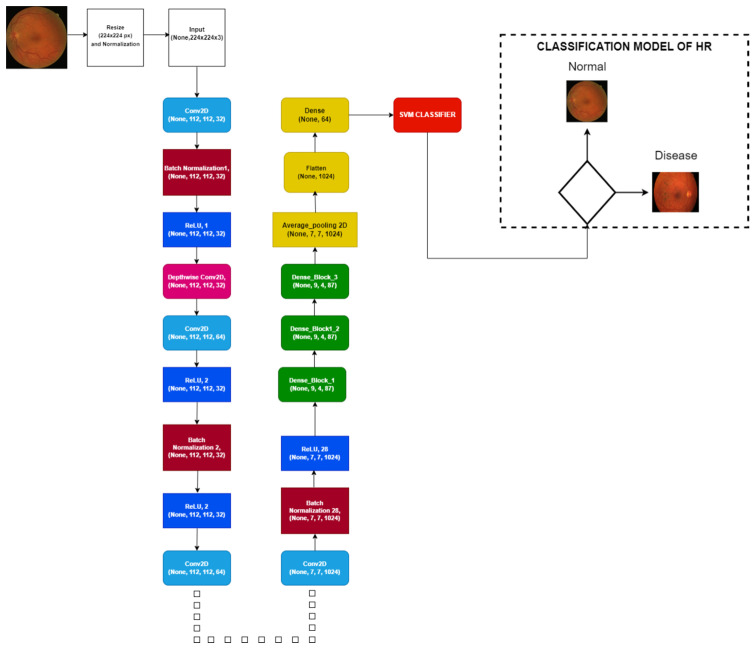
Complete proposed architecture of the Mobile-HR for the classification of hypertensive retinopathy.

**Figure 9 diagnostics-13-01439-f009:**
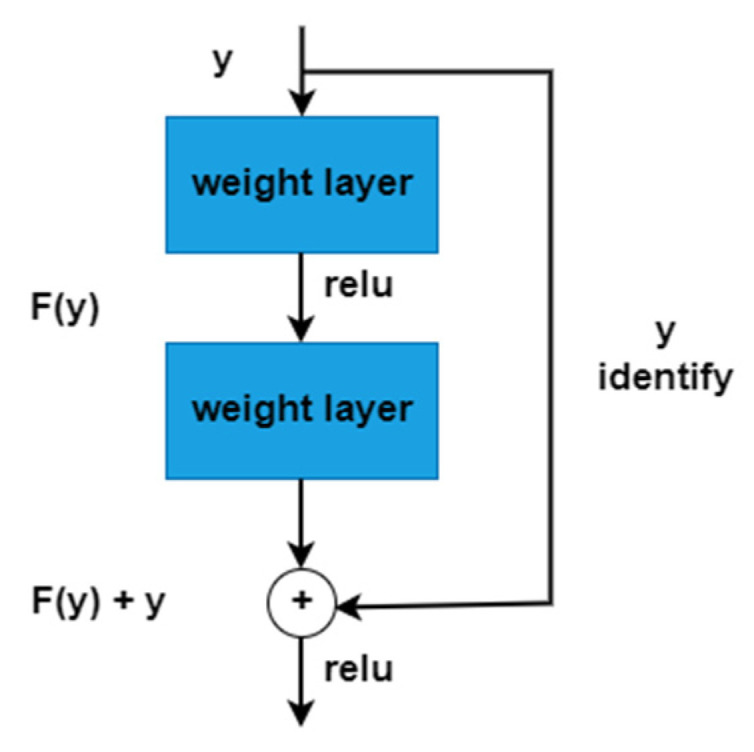
An illustration of how a dense block is used to generate the Mobile-HR model.

**Figure 10 diagnostics-13-01439-f010:**
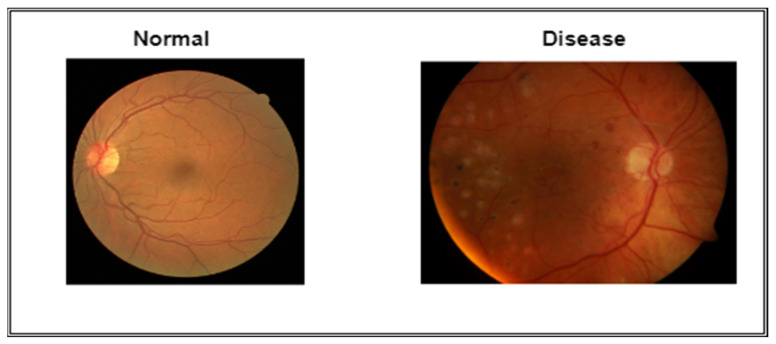
Visual representation of the hypertensive retinopathy images.

**Figure 11 diagnostics-13-01439-f011:**
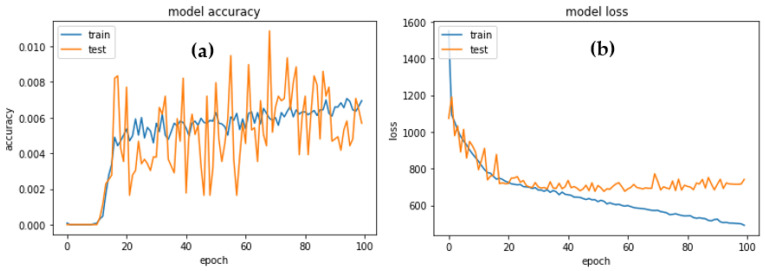
(**a**) Train and validation accuracy; (**b**) train and validation loss graphs of the proposed architecture.

**Figure 12 diagnostics-13-01439-f012:**
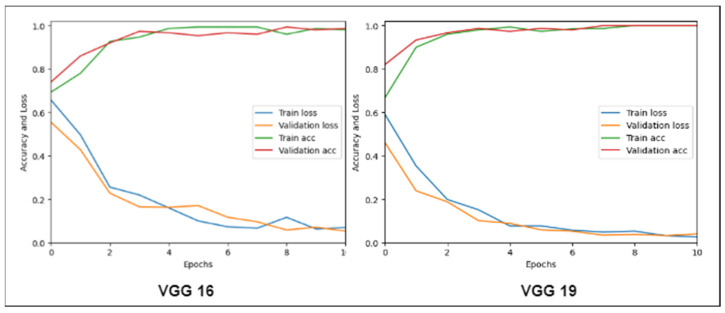
Training validation accuracy and training validation loss of both VGG16 and VGG19.

**Figure 13 diagnostics-13-01439-f013:**
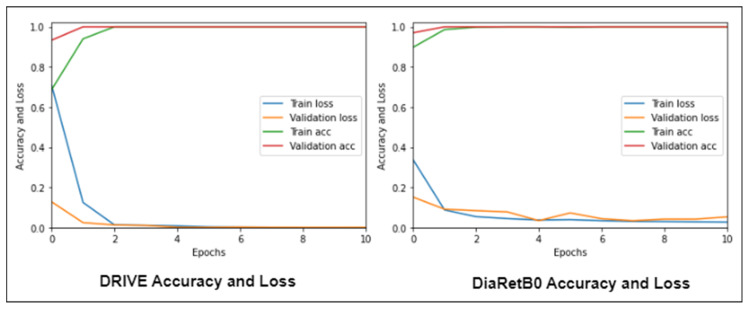
Mobile-HR accuracy and loss using the DRIVE and DiaRetB0 datasets.

**Figure 14 diagnostics-13-01439-f014:**
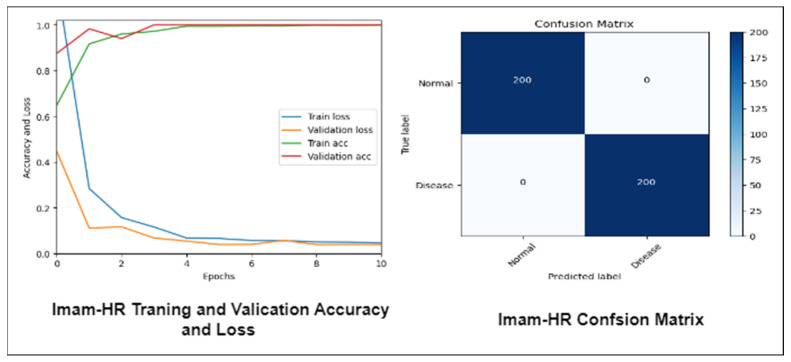
Mobile-HR model’s accuracy, loss, and confusion matrix using the Imam-HR dataset.

**Figure 15 diagnostics-13-01439-f015:**
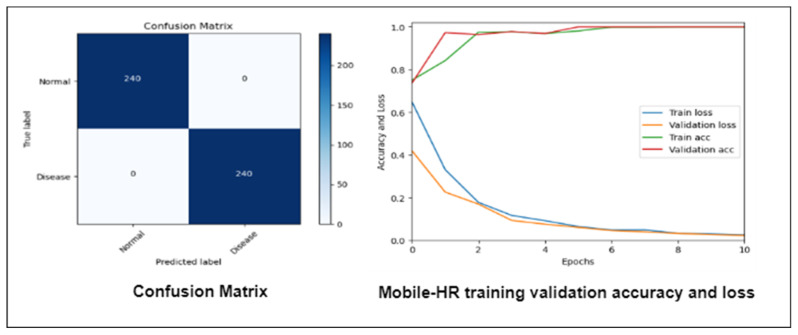
Mobile-HR model’s accuracy, loss, and confusion matrix using the Pak-HR Dataset.

**Figure 16 diagnostics-13-01439-f016:**
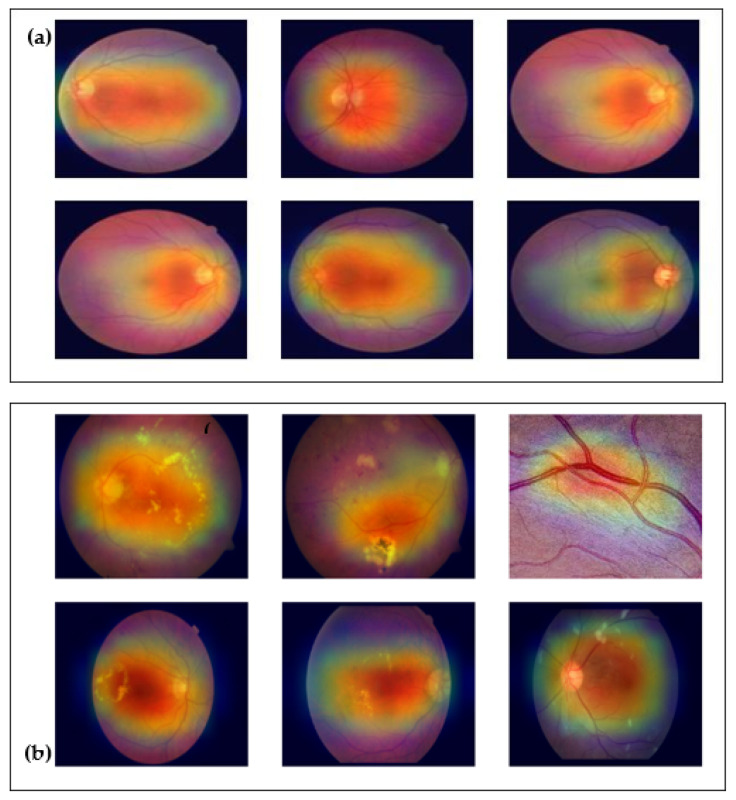
Feature visualization, where Mobile-HR was used to recognize hypertensive retinopathy using heat maps, which highlights the pathologic regions of HR: (**a**) normal heat maps and HR heat maps (**b**) based on the retinographic images.

**Table 1 diagnostics-13-01439-t001:** Seven key factors affected by systematic hypertension.

No.	Key Factors Affected by Systematic Hypertension
1	Hypertension is a major risk factor for cardiovascular disease and is the term used to describe chronically elevated arterial blood pressure.
2	Hypertensive retinopathy (HR), which occurs when blood pressure increases gradually over time or increases abruptly to extremely high levels, can harm the retina.
3	A clinical condition called HR has distinctive fundoscopic symptoms, such as arteriolar narrowing, AV nicking, hard exudates, and cotton wool spots.
4	Evidence of papilledema is used to diagnosis HR (optic disc swelling due to the fact of raised intracranial pressure)
5	Long-term hypertension can cause permanent damage to the lungs, heart, kidneys, and eyes.
6	Ineffective treatment for HR might cause irreparable visual damage. Further retinal vascular disease and the development of diabetic retinopathy (DR) are complications.
7	Malignant hypertension has a 90% death rate after one year if untreated, and delayed treatment may cause irreparable alterations and loss of vision.

**Table 2 diagnostics-13-01439-t002:** Image data of the retina for the Mobile-HR system.

Reference	Name	HR	Non-HR	Images Size	Fundus Images
[27]	DRIVE	100	150	768 × 584 pixels	250
[29]	DiaRetDB0	80	80	1152 × 1500 pixels	160
Private	Pak Eyes Hospitals	2100	3490	1125 × 1264 pixels	5590
Private	Imam-HR	1130	2040	1125 × 1264 pixels	3170
		**3410**	**5760**	Downsizing: 700 × 600 pixels	**9170**

**Table 3 diagnostics-13-01439-t003:** Details of the data and the used image processing.

Techniques for Augmentation	Value
Rotation range	15
Width shift range	0.2
Shear range	0.2
Zoom range	0.2
Crop	Ture
Horizontal flip	True
Vertical flip	False
Fill mood	Nearest
Techniques for augmentation	Values
Rotation range	15

**Table 4 diagnostics-13-01439-t004:** Performance metrics of the Mobile-HR.

Hypertensive Type	SE	SP	ACC	AUC	Error
HR	98%	99%	99%	0.99	0.01
No-HR	99%	98%	99%	0.99	0.01
Average results	99%	99%	99%	0.99	0.01

**Table 5 diagnostics-13-01439-t005:** Performance comparison between Mobile-HR, VGG16, and VGG16.

Methodology	SE	SP	AUC	ACC
VGG19	87%	88%	0.89	91%
VGG16	84%	87%	0.87	90%
Mobile-HR system	99%	99%	0.99	99%

**Table 6 diagnostics-13-01439-t006:** Performance comparison between the Mobile-HR, Triwijoyo-2017, and CAD-HR systems for the diagnosis of HR-related eye disease.

Method	SE	SP	ACC	AUC
Triwijoyo-2017 [29]	78.5%	81.5%	80%	0.84
CAD-HR [30]	94%	96%	95%	0.96
Mobile-HR	99%	99%	99%	0.99

**Table 7 diagnostics-13-01439-t007:** Performance of the CPU/TPU/GPU comparisons of the proposed Mobile-HR model.

Batch	Number of Epochs	* CPU/TPU/GPU (MS)
64	40	300/500/600
128	60	600/500/700
256	80	800/600/750
512	100	950/600/800
1024	120	900/700/800

* MS: milliseconds; CPU: central processing unit; GPU: graphical processing unit; TPU: tensor processing units.

**Table 8 diagnostics-13-01439-t008:** Computational performance of state-of-the-art models with data augmentation and other deep learning techniques compared to the proposed model.

State-of-the-Art Models	Augment	Epochs	Time (S)	ACC	F1-Score
Mobile-HR	Yes	10	2.4	98.1%	98.1%
CNN	Yes	40	12	80.5%	80.5%
AlexNet	Yes	40	17	81.9%	81.9%
MobileNet-LSTM	Yes	40	13	82.3%	82.3%
DenseNet	Yes	40	15	84.8%	84.8%
EfficientNet	Yes	40	18	75.4%	75.4%
CNN-Leaky	Yes	40	20	76.5%	76.5%
DCNN	Yes	40	22	77.9%	77.9%

**Table 9 diagnostics-13-01439-t009:** Limitations of the state-of-the-art-work on hypertensive retinopathy.

Cited	Major Finding	Dataset	Results	Limitation
Sun et al. [17]	To improve the accuracy of the diabetic retinopathy diagnosis model, a convolutional neural network (CNN) model was merged with a batch normalization (BN) layer.	Electronic Record Data = 301 Patients	ACC = 97.56%	Although the proposed model (BNCNN) outperformed logistic regression in terms of accuracy, it still requires further validation on larger datasets with more diverse samples.
Lam et al. [23]	The paper employs convolutional neural networks (CNNs) on color fundus images to perform diabetic retinopathy staging recognition.	Kaggle Dataset = 35,000 Color Fundus Images, Messidor-1 = 1200 Color Fundus Images	SEN = 95%	Errors occurred mostly in misclassifying moderate disease as normal because of the CNNs’ inability to recognize subtle disease signs.
Xu et al. [22]	Exploration and application of deep convolutional neural network methodology for automatic diabetic retinopathy classification utilizing color fundus images.	Kaggle Dataset	ACC = 94%	The study’s dataset was limited, which may restrict its generalizability to larger datasets.
Narayanan et al. [24]	A novel hybrid machine learning architecture for detecting and classifying DR in retinal images.	Asia Pacific Tele-Ophthalmology Society (APTOS) 2019 Dataset = 3662 Retinal Images	ACC = 98.4%	There was no external validation set used to assess their model’s performance.
Hacisoftaoglu et al. [25]	This study uses a DL approach and the ResNet50 network to develop an autonomous detection model for smartphone-based retinal images.	EyePACS = 35,126 Images, Messidor = 1187 Images, Messidor-2 = 1748 Images	ACC = 91% SEN = 92% SPE = 90%	Only a few publicly available datasets were used to train and test the proposed model.
Riaz et al. [26]	The authors analyze retinal images using proposed deep and densely connected networks to distinguish between different stages of diabetic retinopathy.	Messidor-2 = 1748 Images, EyePACS = 35,126 Images	SEN = 98% SPE = 98% SEN = 94% SPE = 97%	Although their technology outperforms existing methods, it still has some false positives, which could lead to unnecessary treatments for those who do not have diabetic retinopathy.
Pavate et al. [27]	The paper uses MobileNet architecture to solve the problem of predicting diabetic retinopathy.	Aptus 2019 Challenge Dataset = 3662 Images	ACC = 95% PRECISION = 95% RECALL = 98% F1-score = 97%	MobileNet is a lightweight and mobile-friendly classifier, but it still requires significant computational power, which may not be available in resource-constrained environments.
Qureshi et al. [28]	The study offers a new computer-aided approach for the early detection and analysis of hypertensive retinopathy, which is connected to high blood pressure.	Imam-HR = 3580 Fundus Image	ACC = 95% SEN = 94% SPE = 96% AUC = 96%	The research does not claim that this approach is generalizable to other datasets.
Abbas et al. [29]	This work presents the development of a novel system called DenseHyper that uses deep residual learning approaches to detect hypertensive retinopathy.	Imam-HR = 4270 Fundus Image	ACC = 95% SEN = 93% SPE = 95% AUC = 96%	The proposed method was tested on limited datasets and may require further validation with larger datasets to ensure its generalizability.
Wu et al. [15]	The proposed transfer learning-based technique for diabetic retinopathy detection can automatically classify DR images with significant value.	Kaggle Dataset = 35,000	ACC = 60%	This study mainly focuses on categorizing DR images into five groups based on the severity of lesions; however, depending on severity levels, there may be more subcategories to examine.
Arsalan et al. [31]	The proposed Vess-Net method for automatic retinal image segmentation is useful in computer-assisted medical image analysis for the identification of disorders, such as hypertension, diabetes and hypertensive retinopathy, and arteriosclerosis.	DRIVE Dataset = 40 RGB Fundus Images	ACC = 96% SEN = 80.2% SPE = 98.1% AUC = 98.2%	It is still a deep learning approach that necessitates large computational resources for training and inference.
Sun et al. [16]	Machine learning methods used to diagnose DR.	Electronic Record Data = 301 Patients	ACC = 92%	This study only focuses on diagnosing diabetic retinopathy via electronic health records (her) data; however, it does not offer any treatment options based on this diagnosis.
Joseph et al. [20]	The research paper discusses how images of the eye, specifically fundus images, can be used to identify medical issues.	Kaggle = 21,000 Fundus Images	ACC = 86%	They still have their own set of restrictions, such as image quality difficulties that limit accuracy.
Arsalan et al. [18]	The study shows a novel approach for the computer-assisted diagnosis of diabetic and hypertensive retinopathy conditions.	DRIVE = 40 Fundus Images, CHASE DB1 = 28 Fundus Images, STARE = 20 Fundus Images	ACC = 82%	Avoiding pre- and post-processing steps can lower system costs, and if used properly, these strategies might even increase segmentation performance.

## Data Availability

The DRIVE dataset is available at: https://drive.grand-challenge.org/DRIVE/ (accessed on 19 March 2019); DiaRetDB0 can be downloaded from: https://www.it.lut.fi/project/imageret/diaretdb0/ (accessed on 19 March 2019); and other datasets are private.

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
