# Peer review of "Mobile-HR: An Ophthalmologic-Based Classification System for Diagnosis of Hypertensive Retinopathy Using Optimized MobileNet Architecture"

_diagnostics, 2023, doi:10.3390/diagnostics13081439_

Round 1

Reviewer 1 Report

The paper „Mobile-HR: A Novel Classification System of Hypertensive Retinopathy Using Optimized MobileNet Architecture“ by Muhammad Zahler Said et al. aims at analysing fundus images to identify the stages and symptoms of hypertensive retinopathy (HR) using machine learning (ML) and deep learning (DL). They developed a HR diagnosis system, Mobile-HR, by integrating a pre-trained model and dense blocks. They reported an accuracy of 99%.  

The authors used a dataset from Pakistani hospitals and internet sources. The analysis was performed with deep characteristics and a color space trained with HR-retina-images. 

Developing a diagnostic tool for HR and improving its accuracy can be useful for medical purposes. 

The following points could be considered for improvement: 

-The introduction contains a general definition of HR. It lacks a detailed medical definition of the condition and the relevance of HR for the patient. Given the fact that the scope of the proposed new method aimed at retinal image analysis a detailed description of ophthalmologic factors would be adequate. 

-After the introduction there is an extensive paragraph of related literature including a table (table 1). This table should better be moved to the discussion. 

-The authors propose a new method of deep learning for HR-diagnosis. The paper is very detailed on the technical side by providing a large amount of information about methods and literature. It could benefit form processing the information in a way, that allowed the audience of a medical journal to follow. 

-In the analysis several medical sources of material were used, like a „huge number of HR retina images“ from Pakistani hospitals. The authors should mention, how they got access to the dataset and if it was ethically approved. 

-The authors report a very high success rate of 99% in detecting HR with their method. Which controls were used to confirm the diagnosis after the proposed method was applied? 

-The title „..classification system…“ is misleading as it does not address actual classification systems of HR in an ophthalmologic sense. 

-As this is an ophthalmologic subject, a professional ophthalmologic contribution to the paper is advisable. 

Author Response

Dear Reviewer,

We have updated the paper according to reviewer #1 and reviewer#2 comments.

Thank you.

Reviewer 2 Report

In this study, the authors proposed methods to detect Hypertensive Retinopathy depended on MobileNet Architectures, and the system achieved 99% accuracy.

From line (228-239), the Figure that described this method not indicated in this paragraph, so please proofreading this paragraph.

In section (3.1. data acquisition), they reduced the images size to (700*600), Why? Or why choose this number of pixels specifically?

Author Response

Original Manuscript ID:  ID: diagnostics-2312777       

Original Article Title: Mobile-HR: An Ophthalmologic-based Classification System for Diagnosis of Hypertensive Retinopathy Using Optimized MobileNet Architecture

To: Editor in Chief,

MDPI, Diagnostics

Re: Response to reviewers

Dear Editor,

Many thanks for insightful comments and suggestions of the referees. Thank you for allowing a resubmission of our manuscript, with an opportunity to address the reviewers’ comments.

We are uploading (a) our point-by-point response to the comments (below) (response to reviewers), (b) an updated manuscript with yellow highlighting indicating changes, and (c) a clean updated manuscript without highlights (PDF main document).

By following reviewers’ comments, we made substantial modifications in our paper to improve its clarity and readability. In our revised paper, we represent the improved manuscript such as:

(1) Revised Abstract, (2) Revised Introduction, (3) Results section, (4) Discussions and Conclusion sections.

We have made the following modifications as desired by the reviewers:

Best regards,

Corresponding Author,

Dr. Qaisar Abbas (On behalf of authors),

Professor.
